# An Empirical Study of Task and Feature Correlations in the Reuse of Pre-trained Models

**Jama Hussein Mohamud**
Mila - Québec Artificial Intelligence Institute
Université de Montréal
`hussein-mohamu.jama@mila.quebec`

**Willie Brink**
Department of Mathematical Sciences
Stellenbosch University
`wbrink@sun.ac.za`

## Abstract

Pre-trained neural networks are commonly used and reused in the machine learning community. Alice trains a model for a particular task, and a part of her neural network is reused by Bob for a different task, often to great effect. To what can we ascribe Bob's success? This paper introduces an experimental setup through which factors contributing to Bob's empirical success could be studied in silico. As a result, we demonstrate that Bob might just be lucky: his task accuracy increases monotonically with the correlation between his task and Alice's. Even when Bob has provably uncorrelated tasks and input features from Alice's pre-trained network, he can achieve significantly better than random performance due to Alice's choice of network and optimizer. When there is little correlation between tasks, only reusing lower pre-trained layers is preferable, and we hypothesize the converse: that the optimal number of retrained layers is indicative of task and feature correlation. Finally, we show in controlled real-world scenarios that Bob can effectively reuse Alice's pre-trained network if there are semantic correlations between his and Alice's task.

## 1 Introduction

Pre-trained models are widely shared in the machine learning community. The weights of the lower layers of a trained neural network are frozen, and the final layer(s) are fine-tuned for use on a new task. This often leads to significant computational savings and better results on the new task, compared to training a model from scratch.

In a fictional setting, Alice trains a deep neural network to classify cats from dogs. She hands her model to Bob, who uses it as a pre-trained model. Bob adapts only the last layer's weights to classify indoors from outdoors scenes, and achieves remarkable accuracy. Is this due to Bob's skills as a machine learning engineer, or is Bob just lucky? If dogs appear in more outdoor scenes, and cats in more indoor scenes, did Alice unknowingly solve most of his problem for him?

In this work, we present an empirical study on how task and feature correlations might affect the success of reusing pre-trained models. Because we don't know the correlation between the features necessary for Alice's and Bob's tasks in real-world scenarios, nor the correlation between their tasks, we create a new experimental computer vision setup with a controllable level of relationship between their tasks. The experimental setup could be seamlessly integrated with AI explainability tools like integrated gradients. By controlling task and feature correlation, we show that Bob's accuracy increases monotonically with the correlation between Alice's and Bob's features and tasks. Furthermore, at zero feature and task correlation Bob can achieve accuracy that is significantly better than random due to Alice's choice of network and initialization. This is particularly evident for convolutional neural networks, which are essentially fully-connected networks with local receptive fields and shared weights.

Our experiments indicate that the optimal selection of layers for Bob to fine-tune depends on the (in practice unknown) correlations between his and Alice's features and tasks. We conjecture that when performance drops dramatically when fine-tuning only later layers of Alice's network, it implies low task correlations. Conversely, if fine-tuning only higher or the final network layer gives good performance, it is likely due to high task and feature correlations. Finally, we show in controlled real-world case studies that when Alice and Bob have semantically correlated tasks, Bob can achieve good performance without fine-tuning any layers on his task.

A number of works study the behaviour of pre-training, for which we give a more comprehensive account in Section 5. Pre-trained transformers with frozen self-attention and feed-forward layers show competitive performance on different modalities (Lu et al., 2021). Architecture depth, model capacity, number of training samples, and initialization influence pre-training (Erhan et al., 2009). Pre-training can also be viewed as a conditioning mechanism for the initialization of a network's weights (Erhan et al., 2010). For networks pre-trained on ImageNet, Kornblith et al. (2019) found a strong correlation between the accuracy of the pre-trained model on ImageNet and the accuracy of a fine-tuned model on another image classification task, but the relationship is sensitive to certain characteristics of that task.

In section 2 we set up a small toy example to illustrate a relationship between correlations in Alice's and Bob's features and transfer accuracy. In section 3 we investigate the effect of correlation in Alice's and Bob's tasks on transfer accuracy, with an image dataset we construct specifically to control task correlation. In section 4 we consider a number of real-world case studies. Section 5 summarizes related works, and section 6 concludes the paper. Code for data generation, model training, experiments and results is available online.[1]

## 2 Feature correlations aid pre-training

We illustrate the value of feature correlations in pre-training and transfer through small generalized linear models with a controllable level $\alpha$ of correlation between the features useful for Alice's and Bob's tasks. With high positively or negatively correlated features, Bob achieves high transfer accuracy. Unsurprisingly, when there is no correlation between the features useful for Alice's and Bob's tasks, Alice ignores Bob's features in her representation, and no transfer is possible.

Let $\boldsymbol{x} \in \mathbb{R}^{2k}$ be a $2k$-dimensional feature vector, for which we generate two binary labels $\boldsymbol{y} = (y^{\text{Alice}}, y^{\text{Bob}})$ as Bernoulli random variables with

$$p(y^{\text{Alice}} = 1|\boldsymbol{x}) = \sigma(\underbrace{[1, \ldots, 1}_{k \text{ copies}}, \underbrace{0, \ldots, 0}_{k \text{ copies}}]^{\mathsf{T}}\boldsymbol{x}), \quad p(y^{\text{Bob}} = 1|\boldsymbol{x}) = \sigma(\underbrace{[0, \ldots, 0}_{k \text{ copies}}, \underbrace{1, \ldots, 1}_{k \text{ copies}}]^{\mathsf{T}}\boldsymbol{x}), \quad (1)$$

where $\sigma(\cdot)$ is the sigmoid function. In the simplest $k = 1$ case Alice's task depends only on $x_1$, like whether there is a cat or dog present in the image, and Bob's task depends only on $x_2$, like whether the image is an indoors or outdoors scene. But because these might be correlated, Alice's network might learn to depend on indoors-outdoors features, thereby helping Bob. We consider two scenarios.

**Scenario 1: pairwise correlated features.** The correlations of distribution $p(\boldsymbol{x}|\alpha)$ are $\boldsymbol{\Sigma}(\alpha) = (1 - \alpha)\boldsymbol{I} + \alpha\boldsymbol{D}$, where $\boldsymbol{I}$ is the identity matrix and $\boldsymbol{D} = \begin{pmatrix} \boldsymbol{I}_k & \boldsymbol{I}_k \\ \boldsymbol{I}_k & \boldsymbol{I}_k \end{pmatrix}$. If $\boldsymbol{b} = [1, \ldots, 1, 0, \ldots, 0]^{\mathsf{T}}$ indicates the binary vector in equation 1, then the scalar $a = \boldsymbol{b}^{\mathsf{T}}\boldsymbol{x} \sim \mathcal{N}(0, k)$.

**Scenario 2: globally correlated features.** In a more realistic scenario, Alice's and Bob's features are internally correlated through $\boldsymbol{\Sigma}(\alpha) = (1 - \alpha)\boldsymbol{I} + \boldsymbol{D}(\alpha)$, where $\boldsymbol{D}(\alpha) = \begin{pmatrix} |\alpha|\mathbf{1}_k & \alpha\mathbf{1}_k \\ \alpha\mathbf{1}_k & |\alpha|\mathbf{1}_k \end{pmatrix}$ with $\mathbf{1}_k$ a $k \times k$ all-ones matrix. It mimics cases where groups of pixels are correlated. Here, $a = \boldsymbol{b}^{\mathsf{T}}\boldsymbol{x} \sim \mathcal{N}(0, (1 - \alpha)k + |\alpha|k^2)$, and the standard deviation grows with $k$ and not $\sqrt{k}$.

**A rudimentary network.** Alice trains a simple two-layer neural network with first-layer parameters $\boldsymbol{w} \in \mathbb{R}^{2k}$ and a scalar parameter $v$ in the second layer to predict $y^{\text{Alice}}$. It has a functional form

$$p_{\boldsymbol{w},v}(y^{\text{Alice}} = 1|\boldsymbol{x}) = \sigma(v \cdot \text{identity}(\boldsymbol{w}^{\mathsf{T}}\boldsymbol{x})) \quad (2)$$

---

[1] https://github.com/engmubarak48/feature-task-correlations

with identity$(z) = z$. Without loss of generality Alice clamps $v^* = 1$. She minimizes the empirical cross entropy loss on samples from $p(y^{\text{Alice}}|\boldsymbol{x})p(\boldsymbol{x}|\alpha)$, with an $L_2$ regularizer[2] $\lambda\|\boldsymbol{w}\|^2$, to obtain $(\boldsymbol{w}^*, v^*)$. Alice then freezes the "pretrained" weights $\boldsymbol{w}^*$, which she hands over to Bob. Bob uses the "backbone" $f(\boldsymbol{x}) = \text{identity}(\boldsymbol{w}^{*\mathsf{T}}\boldsymbol{x})$ in his network, $p_{v'}(y^{\text{Bob}} = 1|\boldsymbol{x}) = \sigma(v'f(\boldsymbol{x}))$, to predict $y^{\text{Bob}}$. He similarly minimizes the empirical cross entropy loss on samples from $p(y^{\text{Bob}}|\boldsymbol{x})p(\boldsymbol{x}|\alpha)$ only over his last layer's parameter $v'$.

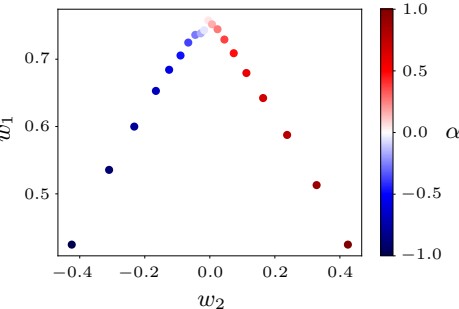

Figure 1: Alice's weights $\boldsymbol{w}^*$ in equation 2 as a function of $\alpha$. At $\alpha = 0$ Alice recovers $\boldsymbol{w}^* \propto [1, 0]$, but as $|\alpha| \to 1$ she incorporates more of Bob's feature $x_2$ until $\boldsymbol{w}^* \propto [1, 1]$ or $\boldsymbol{w}^* \propto [1, -1]$.

Alice's weights as a function of $\alpha$, for $k = 1$, are shown in figure 1. The trivial illustration serves to highlight that Alice does learn a weight $w_2$ that is useful for Bob's task, simply because $x_2$ is correlated with the feature $x_1$ that she requires.

Figures 2(a) and 2(b) show Bob's classification accuracy as a function of the feature correlation parameter $\alpha$, for Scenario 1. Figure 2(a) shows Bob's accuracy "as is", i.e. using Alice's $(\boldsymbol{w}^*, v^*)$. When features are negatively correlated, Bob's accuracy is worse than a random guess. In figure 2(b) Bob has recourse to fine-tune a last layer; he keeps Alice's $\boldsymbol{w}^*$ and optimizes for $v$ in equation 2, and benefits from negative correlations and Alice's model using the "absence" of his features. Figures 2(c) and 2(d) illustrate Scenario 2, again without and with fine-tuning. The $k$ features required for Bob's task need only be very weakly correlated with those of Alice. As long as there are enough such features, Bob can rely on Alice's model.

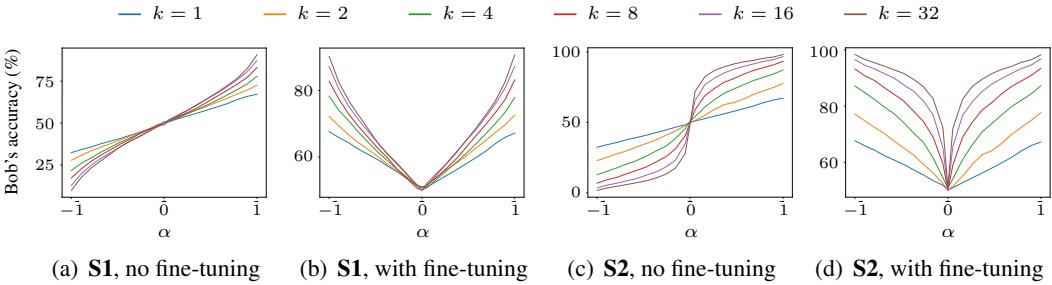

(a) **S1**, no fine-tuning    (b) **S1**, with fine-tuning    (c) **S2**, no fine-tuning    (d) **S2**, with fine-tuning

Figure 2: Bob's accuracy for Scenario 1 (**S1**) and Scenario 2 (**S2**), when he uses Alice's network "as is" with no fine-tuning (Alice's $\boldsymbol{w}^*$ and $v^*$ in equation 2), and when he uses Alice's $\boldsymbol{w}^*$ and fine-tunes $v$ for his task. $k$ is the number of input features tied to Alice's task and to Bob's task.

In this small generalized linear model, zero-correlated features make Alice's pre-trained model useless for Bob, as she removes all information that is irrelevant for her. Against intuition, this statement is *not* true for deeper networks due to optimization artefacts, or convolutional networks due to shared weights. In deeper networks, "distractor features" could be carried to the last layer, allowing Bob to perform well. We investigate this further in the next section, through the construction of image datasets with carefully controlled levels of correlation between Alice and Bob's tasks.

## 3    Task correlations aid pre-training

To examine the impact of feature and task correlations on the success of pre-training in deeper networks, we introduce a setup in which these correlations could be controlled experimentally. This section then examines Bob's success in using Alice's pre-trained network as a function of task coupling, network type, and number of fine-tuned layers.

---

[2]The regularizer is added so that Alice's solutions are unique, for instance when $\alpha = 1$ and $x_1 = x_2$.

### 3.1 Constructing image data with controlled correlation between tasks

Input images are generated through horizontal concatenation of samples from two domains: $\mathcal{D}^{\text{left}}$ and $\mathcal{D}^{\text{right}}$. Alice sees the full concatenated images and only the left side labels. Her trained model is handed over to Bob, who sees the full images and only the right side labels. We control the level of correlation between Alice's and Bob's task labels through a parameter $\beta \in [0, 1]$, and generate samples according to algorithm 1. $\beta = 0$ would lead to zero correlation between Alice's and Bob's tasks, where Alice's neural network should ignore the input features for Bob's task, while $\beta = 1$ would lead to perfect correlation, where Alice's neural network would benefit from the input features for Bob's task.

For $\mathcal{D}^{\text{left}}$ and $\mathcal{D}^{\text{right}}$ we consider four separate combinations of the MNIST (Deng, 2012) and Street View House Numbers (SVHN) (Netzer et al., 2011) datasets: MNIST-MNIST, MNIST-SVHN, SVHN-MNIST and SVHN-SVHN. We rescale the $28 \times 28$ grayscale MNIST images to match the $32 \times 32$ color SVHN images, such that the concatenated images are each of shape $(32, 64, 3)$. Figure 3 shows a sample from the MNIST-MNIST set in (a), and one from MNIST-SVHN in (b).

---

**Algorithm 1** Sampling from $p_\beta(\boldsymbol{x}, y^{\text{Alice}}, y^{\text{Bob}})$

1: **input:** $\beta \in [0, 1]$, $\mathcal{D}^{\text{left}} = \{\boldsymbol{x}_i^{\text{l}}, y_i^{\text{l}}\}_{i=1}^{I}$, $\mathcal{D}^{\text{right}} = \{\boldsymbol{x}_j^{\text{r}}, y_j^{\text{r}}\}_{j=1}^{J}$
2: **while** true **do**
3: $\quad i \sim \text{unif}\{1, \ldots, I\}$        # random sample from $\mathcal{D}^{\text{left}}$
4: $\quad y^{\text{Alice}} \leftarrow y_i^{\text{l}}$        # Alice's task label
5: $\quad u \sim \text{unif}[0, 1]$
6: $\quad$ **if** $u < \beta$ **then**        # sample from $\mathcal{D}^{\text{right}}$
7: $\quad\quad j' \sim \text{unif}\{j : y_j^{\text{r}} = y^{\text{Alice}}\}$
8: $\quad$ **else**
9: $\quad\quad j' \sim \text{unif}\{1, \ldots, J\}$
10: $\quad$ **end if**
11: $\quad y^{\text{Bob}} \leftarrow y_{j'}^{\text{r}}$        # Bob's task label
12: $\quad \boldsymbol{x} = \text{concat}(\boldsymbol{x}_i^{\text{l}}, \boldsymbol{x}_{j'}^{\text{r}})$        # the concatenated image
13: $\quad$ **yield** $\boldsymbol{x}, y^{\text{Alice}}, y^{\text{Bob}}$
14: **end while**

---

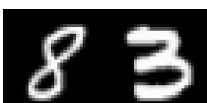

(a) $y^{\text{Alice}} = 8$, $y^{\text{Bob}} = 3$

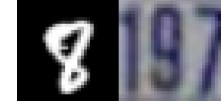

(b) $y^{\text{Alice}} = 8$, $y^{\text{Bob}} = 9$

Figure 3: Two example images $\boldsymbol{x}$ with their task labels. Both used MNIST for $\mathcal{D}^{\text{left}}$, and (a) MNIST and (b) SVHN for $\mathcal{D}^{\text{right}}$.

### 3.2 Experimental setup

Given data from $p_\beta(\boldsymbol{x}, y^{\text{Alice}}, y^{\text{Bob}})$, Alice trains a network with $m$ hidden layers:

$$p_{\boldsymbol{W}_{1:m}, \boldsymbol{V}}(y^{\text{Alice}} = y | \boldsymbol{x}) = s_{\boldsymbol{V}}\big(f_{\boldsymbol{W}_{1:m}}(\boldsymbol{x}); y\big), \tag{3}$$

where $s_{\boldsymbol{V}}(\cdot; y)$ denotes the output probability for class $y$, from an output layer with parameters $\boldsymbol{V}$ and softmax activation, and $f_{\boldsymbol{W}_{1:m}}$ is a composite neural network function. Given Alice's trained parameters $\boldsymbol{W}_{1:m}^*$, Bob either adapts only the output layer parameters $\boldsymbol{V}'$ to predict $y^{\text{Bob}}$:

$$p_{\boldsymbol{W}_{1:m}^*, \boldsymbol{V}'}(y^{\text{Bob}} = y | \boldsymbol{x}) = s_{\boldsymbol{V}'}\big(f_{\boldsymbol{W}_{1:m}^*}(\boldsymbol{x}); y\big), \tag{4}$$

or freezes Alice's parameters $\boldsymbol{W}_{1:\ell-1}^*$ and retrains both $\boldsymbol{W}_{\ell:m}'$ and $\boldsymbol{V}'$:

$$p_{\boldsymbol{W}_{1:\ell-1}^*, \boldsymbol{W}_{\ell:m}', \boldsymbol{V}'}(y^{\text{Bob}} = y | \boldsymbol{x}) = s_{\boldsymbol{V}'}\big(f_{\boldsymbol{W}_{1:\ell-1}^*, \boldsymbol{W}_{\ell:m}'}(\boldsymbol{x}); y\big). \tag{5}$$

We consider two network architectures. The first is a **fully-connected network** where the input is flattened to a 6144-dimensional vector and passed through two hidden layers with 1024 and 512 nodes respectively, and an output softmax layer with 10 nodes. The hidden layers have batch normalization and ReLU activation, and the first also dropout with rate 0.25. The second architecture is a **convolutional network** containing six convolutional layers with batch normalization and ReLU, two fully-connected ReLU layers, and a softmax output layer. The even-numbered convolutional blocks have $2 \times 2$ max pooling and dropout (0.25), and there is also dropout (0.25) after the first fully-connected layer. The convolutional layers have 32, 64, 128, 256, 512 and 1024 filters, respectively, all of size $3 \times 3$, and the fully-connected layers have 1024, 512 and 10 nodes, respectively.

For each of the four combinations of MNIST and SVHN we sample $60,000$ images for testing and *resample* $60,000$ images for each epoch in training. Alice's networks are trained with stochastic gradient descent for 15 epochs, using a learning rate of $0.001$, momentum of $0.9$, weight decay of $5 \times 10^{-4}$ and a mini-batch size of 256. Bob's fine-tuning is similar, except for a cosine learning rate scheduler from $0.3$ to $0$ over 10 epochs.

### 3.3 Results

Figure 4 shows Bob's test accuracy after fine-tuning the parameters of the output layer to predict $y^{\mathrm{Bob}}$, following equation 4, for fully-connected and convolutional architectures. As expected, Bob's accuracy improves as the task correlation parameter $\beta$ increases. At low values for $\beta$, Bob benefits more if Alice chose a convolutional network architecture. Curiously, we might expect Bob's network to perform no better than random at $\beta = 0$, i.e. a 10% accuracy, but this is not the case. Alice initialized her network with random weights, and even after training, her weights encode *some* of Bob's input features that are irrelevant for her task. We show in section 3.5 that this seems unavoidable unless Alice has an oracle's insight into feature selection.

Instead of fine-tuning only the output layer, Bob could choose to fine-tune earlier layers of Alice's network as well, following equation 5. Figure 5 shows Bob's test accuracy when he freezes the first $\ell - 1$ trainable layers of Alice's network and fine-tunes layers $\ell$ to $m + 1$ (recall that Alice's network has $m$ hidden layers plus an output layer), with $\ell = m + 1$ corresponding to the situation in figure 4. Fine-tuning more layers improves Bob's accuracy. This improvement is less pronounced at higher levels of correlation between Alice's and Bob's tasks, implying that the weights in deeper layers of Alice's pre-trained network are more useful for Bob when the tasks are more correlated.

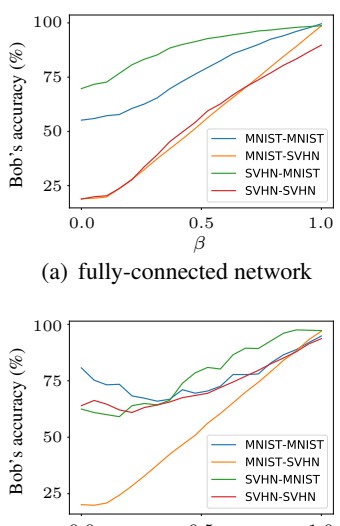

(a) fully-connected network

(b) convolutional network

Figure 4: Bob's accuracy after fine-tuning only the output layer of Alice's networks for his task (equation 4), as functions of $\beta$ and different choices of $\mathcal{D}^{\mathrm{left}}$ and $\mathcal{D}^{\mathrm{right}}$.

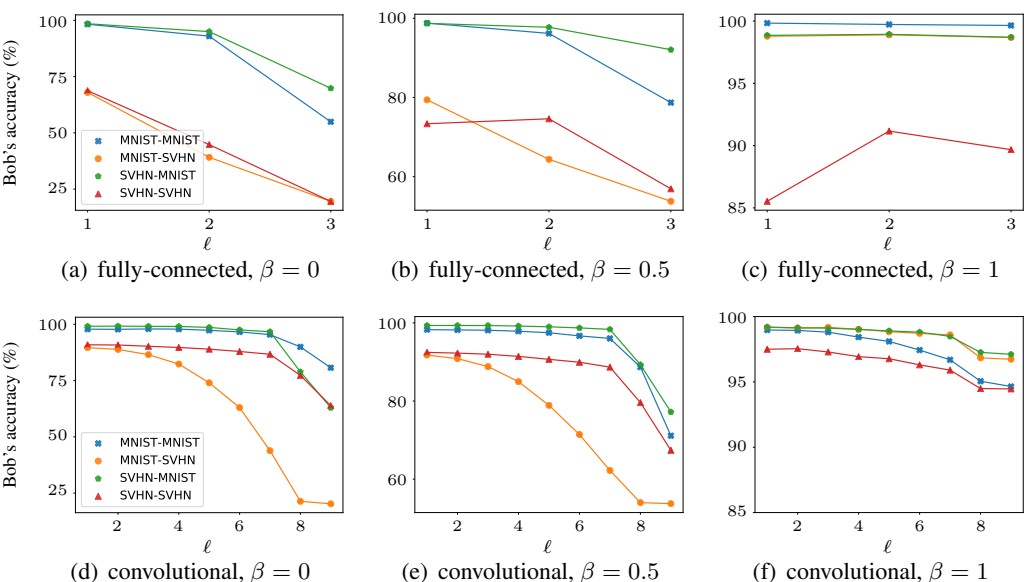

(a) fully-connected, $\beta = 0$    (b) fully-connected, $\beta = 0.5$    (c) fully-connected, $\beta = 1$

(d) convolutional, $\beta = 0$    (e) convolutional, $\beta = 0.5$    (f) convolutional, $\beta = 1$

Figure 5: Bob's accuracy after freezing the first $\ell - 1$ layers of Alice's pre-trained network and fine-tuning layer $\ell$ to the output (equation 5), for the fully-connected and convolutional networks, and for different values of the task correlation parameter $\beta$.

## 3.4 Input attribution

For an indication of what Alice's and Bob's networks might be focusing on when classifying images, we make use of the attribution method of integrated gradients (Sundararajan et al., 2017) with a black image as baseline. For simplicity we focus on the *convolutional* networks, and the case where Bob fine-tunes only the output layer of Alice's pre-trained network.

Figure 6 shows integrated gradients for Alice's and Bob's networks, for sample test images from the four combinations of MNIST and SVHN. In the case of zero task correlation ($\beta = 0$), Alice's network has learned to almost completely ignore the right side of the image which contains features that are only useful for Bob's task. Bob's fine-tuned network does pick up features from the right side, but also retains most of Alice's features (likely due to Bob's limitations in adapting the network for his task). When there is perfect task correlation ($\beta = 1$), Alice's network has learned to pay more attention to Bob's features. We see this especially for (a) MNIST-MNIST and (c) SVHN-MNIST, where there is clear and useful signal for Alice's task in the right side of the images.

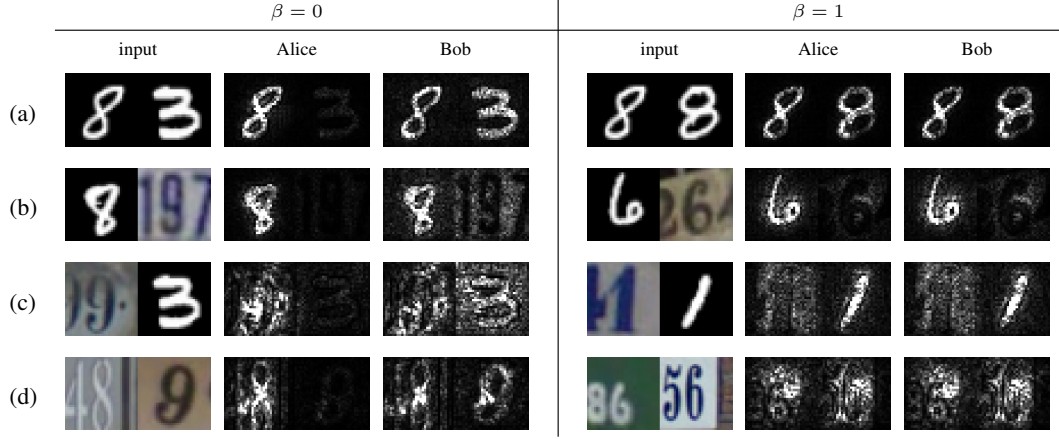

Figure 6: Integrated gradients computed for Alice's and Bob's convolutional networks, where Bob fine-tuned only the output layer, on sample inputs from (a) MNIST-MNIST, (b) MNIST-SVHN, (c) SVHN-MNIST, and (d) SVHN-SVHN, for zero ($\beta = 0$) and perfect ($\beta = 1$) task correlation.

We also calculated integrated gradients on $1,000$ images sampled from each of the datasets, computed the average integrated gradient value of the left side and right side of each sample, and plotted histograms of these averages. These are shown in figure 7. The overlapping histograms for Alice's network imply that Alice found use in both her and Bob's features when $\beta = 1$, while the separated histograms for Alice would imply that she used mostly only her own features when $\beta = 0$. The overlapping histograms for Bob's network suggest strong reliance on Alice's features, even when their tasks are uncorrelated ($\beta = 0$).

## 3.5 Oracle initialization

In the small experiments in section 2, we showed that at zero feature correlation ($\alpha = 0$), Alice assigns zero weights to Bob's features and Bob's accuracy is no better than random. However, in section 3.3, we showed that depending on $p_0(\boldsymbol{x}, y^{\text{Alice}}, y^{\text{Bob}})$, and Alice's and Bob's network architecture and how many layers Bob fine-tunes, Alice does not blank Bob's features and Bob might get strong performance at $\beta = 0$.

What if Alice had access to an oracle and knew which image pixels are uncorrelated to her task? For simplicity, using *fully-connected* networks, we perform experiments where we initialize Alice's weights corresponding to Bob's features either randomly or to zero. In the integrated gradients in figure 8 we see that with random initialization, Alice uses Bob's irrelevant features when both $\beta = 0$ and $\beta = 1$. When Alice initializes weights corresponding to Bob's features to zero she effectively wipes Bob's features at $\beta = 0$ and achieves better generalization performance, as indicated in table 1. It should be noted that when trained with stochastic gradient descent, Alice's zero-initialized weights are perturbed slightly, allowing Bob to perform better than random on his task. After fine-tuning

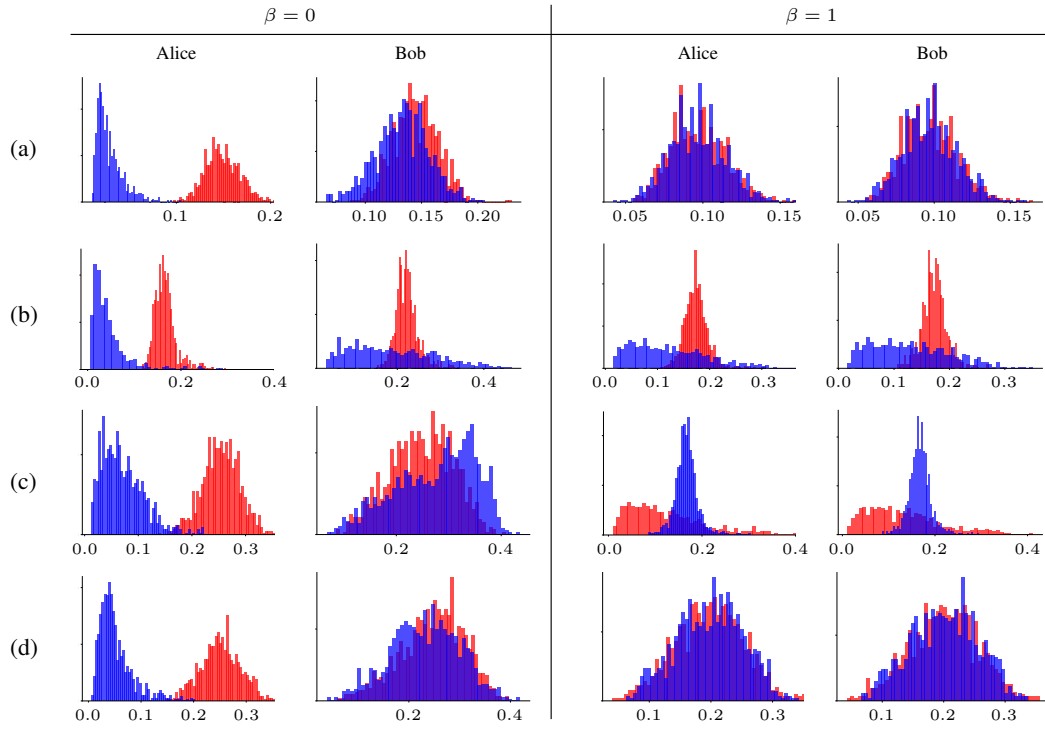

Figure 7: Normalized histograms over $1,000$ samples of (a) MNIST-MNIST, (b) MNIST-SVHN, (c) SVHN-MNIST and (d) SVHN-SVHN, of average integrated gradient responses over the **left** and **right** sides of input images, for Alice's and Bob's convolutional networks (where Bob fine-tuned the output layer).

the output layer Bob achieves test accuracies of about 49, 20, 69 and 19% on MNIST-MNIST, MNIST-SVHN, SVHN-MNIST and SVHN-SVHN, respectively. This is similar to his performance in figure 4(a) at $\beta = 0$, where Alice did not initialize any weights to zero.

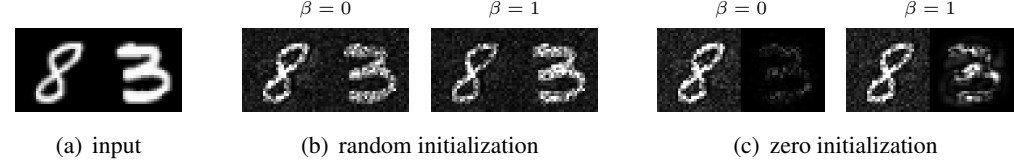

Figure 8: Integrated gradients for Alice's fully-connected network on a sample from MNIST-MNIST, for $\beta = 0$ and $\beta = 1$. When initialized randomly her network makes use of Bob's features, regardless of the level of correlation in their tasks. When she initializes the weights corresponding to Bob's features to zero, her network ignores most of Bob's features at $\beta = 0$.

Table 1: Test accuracies obtained by Alice with her fully-connected network, on the four datasets with $\beta = 0$ and $\beta = 1$. We compare her accuracy when weights corresponding to Bob's input features are initialized randomly and when those weights are initialized to zero.

| $\beta$ | MNIST-MNIST | | MNIST-SVHN | | SVHN-MNIST | | SVHN-SVHN | |
|---|---|---|---|---|---|---|---|---|
| | random | zero | random | zero | random | zero | random | zero |
| 0 | 97.76 | **98.13** | 97.85 | **97.99** | 77.74 | **79.43** | 78.89 | **79.99** |
| 1 | 99.53 | **99.59** | **98.28** | 98.18 | 98.41 | **98.55** | **88.24** | 88.03 |

# 4 Real-world image data

We continue our empirical study on possible effects of task correlation on the reuse of pre-trained models, now using real-world image datasets. As before, Alice trains a network on some dataset and Bob fine-tunes the output layer of Alice's pre-trained network for his own task. We consider one binary image classification task (and dataset) for Alice, and four separate tasks (and datasets) for Bob with varying levels of correlation to Alice's task. Here the level of correlation will be more speculative compared to previous sections where we could control correlation finely. Nevertheless, we show that Bob's success seems to depend on semantic and contextual relationships between his and Alice's tasks.

## 4.1 Alice's and Bob's datasets and tasks

By common sense we speculate that certain visual similarities in the image datasets for two different tasks may imply a level of correlation between those tasks. Alice trains an image classification model on Kaggle's "`dog` and `cat`" dataset.[3] It contains $20,000$ training images and $5,000$ test images. For Bob's task we sample four separate subsets of Open Images V4 (Kuznetsova et al., 2020):

| | | | |
|---|---|---|---|
| set 1 | `camel` and `bed` | $1,348$ training images | $338$ test images |
| set 2 | `coffee` and `coin` | $2,296$ training images | $574$ test images |
| set 3 | `cattle` and `bed` | $3,680$ training images | $920$ test images |
| set 4 | `coffee` and `bread` | $3,680$ training images | $920$ test images |

These datasets are all class-balanced. Bob's task on set 1 may have a significant level of correlation with Alice's "`dog` and `cat`" task, since it could well be that images of dogs and camels are taken mostly outdoors, while images of cats and beds are taken mostly indoors. By contrast, Bob's task on set 2 may have little correlation with Alice's task. Bob's tasks on sets 3 and 4 may also give similarly high and low levels of correlation with Alice's task, respectively, and these sets have more samples compared to sets 1 and 2.

Alice trains a convolutional neural network with the ResNet-18 architecture (He et al., 2016) on her "`dog` and `cat`" dataset for 200 epochs, and achieves a test accuracy of around 94%. Bob then uses Alice's pre-trained network for each of his four tasks. We consider the case where Bob uses Alice's network "as is" with no fine-tuning, and also the case where Bob fine-tunes the weights in the output layer for his task.

## 4.2 Results

Figure 9 shows Bob's test accuracy on each of his four tasks, with no fine-tuning and also after fine-tuning the weights in the output layer. Without fine-tuning, Bob achieves remarkable transfer

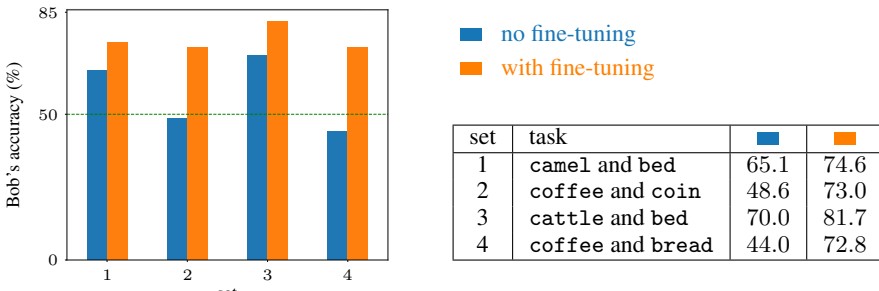

| set | task | ■ | ■ |
|---|---|---|---|
| 1 | `camel` and `bed` | 65.1 | 74.6 |
| 2 | `coffee` and `coin` | 48.6 | 73.0 |
| 3 | `cattle` and `bed` | 70.0 | 81.7 |
| 4 | `coffee` and `bread` | 44.0 | 72.8 |

Figure 9: Bob's test accuracy before and after fine-tuning, on each of his four tasks. If his task is correlated to Alice's "`dog` and `cat`" task, Bob gets high accuracy even without fine-tuning (sets 1 and 3). For uncorrelated tasks, Bob gets improvements only after fine-tuning on his task (sets 2 and 4).

---

[3]`https://www.kaggle.com/competitions/dogs-vs-cats/data`

accuracy on the "`camel` and `bed`" and "`cattle` and `bed`" tasks. In both cases fine-tuning improves his accuracy. For the two tasks of Bob that are not correlated to Alice's task, namely "`coffee` and `coin`" and "`coffee` and `bread`", Bob does slightly worse than random with no fine-tuning of Alice's network. However, when he fine-tunes the output layer his accuracy improves. A similar trend was observed in section 3, where fine-tuning seems to improve transfer accuracy regardless of the level of task correlation.

The results suggest that a greater boost in performance can be expected from fine-tuning when there is less correlation in Alice's and Bob's tasks (compare the improvements for sets 1 and 3 with those for sets 2 and 4). The results further suggest that when Bob uses a larger dataset, he can expect a greater performance boost from fine-tuning (compare the improvements for sets 1 and 2 with those for sets 3 and 4).

## 5    Related work

Development and use of pre-trained models have seen tremendous growth over the last decade, due to the proliferation of data and compute resources, and the clear benefits of transfer learning in data-scarce environments. Today, pre-trained models find use in various downstream tasks that may differ significantly from the tasks those models were initially trained on. There are many examples of popular pre-trained models for computer vision (Krizhevsky et al., 2012; Simonyan & Zisserman, 2014) and natural language processing (Vaswani et al., 2017; Liu et al., 2019; Yang et al., 2019). Recently, Lu et al. (2021) showed that pre-training on language can improve performance and compute efficiency even on different downstream modalities, such as vision and protein fold prediction.

Despite widespread use of pre-trained models, questions on precisely how and why pre-training and transfer learning work remain largely unanswered. It is known that pre-training can be useful as an initialization strategy to aid optimization on the a task (Bengio et al., 2007). It is also known that fine-tuning more layers of a pre-trained model can improve transfer accuracy, but there are limits (Erhan et al., 2009). Related to our work, Neyshabur et al. (2020) found through experimental analysis that some of the benefits of transfer learning comes from the pre-trained network learning low-level statistics of the input data. For models pre-trained on ImageNet, Kornblith et al. (2019) found strong correlation between ImageNet accuracy and transfer accuracy. But there is also evidence supporting our claims that transfer accuracy can be dependent on task characteristics. Raghu et al. (2019) showed that smaller and simpler convolutional architectures could outperform state-of-the-art ImageNet models on certain medical image applications.

Building large pre-trained models for widespread use can be costly in terms of time, money, compute resources, and data collection. Yet the extent of their reusability may not always be clear. Therefore, understanding how pre-training and transfer learning work is of paramount importance. Our work begins to shed light on a new perspective on the role of task and feature correlation in the reuse of pre-trained models.

## 6    Conclusion and future work

We investigated possible effects of feature and task correlations on the performance of pre-training and transfer learning. We set up small experiments to verify that zero correlation between the features used by Alice (the pre-trainer) and by Bob (the user) led to unsuccessful transfer in a simple, shallow network. However, with only weak feature correlation Bob is able to find use in Alice's weights, especially after fine-tuning.

We then introduced an algorithm to construct synthetic image datasets with full control over the correlation between Alice's and Bob's tasks. We saw a clear increase in Bob's accuracy with the correlation parameter, in both fully-connected and convolutional architectures. We also found that pre-trained weights in deeper layers of Alice's network might be more useful for Bob when their tasks are more correlated. Through an analysis of integrated gradients, we found that Alice's network would learn to ignore input features that are not correlated to her task, making it difficult for Bob to retrieve those features should he require them for his task. Through fine-tuning Bob is able to recover features completely wiped by Alice, to some degree.

Finally, we demonstrated the potential applicability of some of our findings on real-world datasets, through a few experiments on carefully chosen datasets for which we could postulate visual similarity and contextual correlation.

Constructing sensible datasets where we can easily and precisely control feature and task correlations between Alice and Bob, remains a challenge for future work. Extensions to other architectures and data modalities can also be investigated.

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
