# OpenReview forum: "An Empirical Study of Task and Feature Correlations in the Reuse of Pre-trained Models"
_NeurIPS.cc/2025/Workshop/UniReps — UniReps2025_

### Official Review · Reviewer_8DSL · 2025-09-12

**Confidence:** 4

**Review:**

**Summary:**

This paper investigates how task and feature correlations affect the success of reusing pre-trained models. Using both synthetic setups with controllable correlations and real-world case studies, the authors first show that transfer performance improves monotonically with task/feature correlation. They further demonstrate that performance significantly better than random can arise even at zero task/feature correlation. The paper also includes additional analyses such as layer-wise fine-tuning experiments, integrated gradients for attribution, and oracle initialization studies.

**Strengths And Weaknesses:**

**Strengths**
- The paper is written in a clear and accessible manner, with well-motivated toy examples leading into more complex experiments.
- The paper explores task and feature correlations in the reuse of pre-trained models from diverse experimental perspectives, including controllable correlation setups, layer-wise fine-tuning experiments, attribution via integrated gradients, and oracle initialization studies.

**Weaknesses**
- The paper lacks a unified definition or measurement of feature/task correlation. In Section 2, when Alice and Bob receive different labels for the same input, feature correlation is defined as correlations between different input dimensions. In Section 3, under the same input with different labels, task correlation is defined as label correlations. In Section 4, Alice and Bob's inputs come from different datasets, and task correlation is defined by semantic relationships between classification tasks. The paper does not clearly clarify how these three notions are consistent with each other. Since task/feature correlation is the central explanatory factor of the work, the absence of a unified definition limits the comparability of results across new tasks.
- The current conclusions about correlations may overlook other factors that affect Bob's test accuracy. Two examples are:
   - In Section 3, even when Alice's and Bob's task correlation parameter $\beta$ is fixed, differences between $\mathcal{D}^{\text{left}}$ and $\mathcal{D}^{\text{right}}$ domains can influence Bob’s accuracy. For example, Figure 6 suggests that when $\beta=1$, Alice finds features in $\mathcal{D}^{\text{left}}$ and $\mathcal{D}^{\text{right}}$ similarly learnable. However, if one domain contains certain features that are much easier to learn as shortcuts, then Bob's performance could diverge.
   - In Section 4.1, the authors argue that Alice's and Bob's tasks are similar because their relationships between background features and labels are alike. However, Alice's reliance on spurious correlations between backgrounds and labels  could also affect Bob's performance. For example, even if Bob's task is semantically close (e.g., cats vs. dogs), if most of Bob's dogs appear indoors and most of his cats outdoors, the accuracy may be reduced.

**Score:**

3

**Topic Fit:**

2

---

### Official Review · Reviewer_F1jV · 2025-09-12
**Some interesting findings and goals, but conceptual misunderstanding led to flawed methodological choices**

**Confidence:** 4

**Review:**

Summary:

The paper investigates the relationship between the feature correlation of pairs of tasks and the effectiveness of transfer learning. The authors introduce a feature vector x with two possible binary Bernoulli random variable labels under the pairwise and globally correlated features scenarios. The authors conduct experiments across a range of architectures, including ridge-regularized logistic classifiers with scaled inputs under the activation function, fully connected and convolutional networks trained on a concatenated MNIST + SVHN dataset, and image classification models pre-trained on the Cats vs. Dogs dataset. The findings suggest that pretrained networks tend to suppress uncorrelated feature vectors, although in larger architectures, some of these features persist. Fine-tuning can recover a significant fraction of the lost features. They also conduct fine-tuning experiments on four real-world image datasets with varying degrees of visual correlation.

Strengths:

The mathematical analysis of feature correlations is sound despite simplifications and weaknesses in empirical validation.

The authors introduce an interesting use case of integrated gradients for visualizing the effects of feature initialization.

The authors note that the perturbation effect of stochastic gradient descent (SGD) on the zero-initialized weights leads to an improvement in performance.

Weaknesses

The section on the rudimentary network appears unnecessarily elaborate for what is essentially a discussion of a logistic classifier. By clamping v*=1, Alice’s model reduces to a standard logistic classifier, while Bob’s model amounts to a simple rescaling of the input to the sigmoid function. While not explicitly expressed in terms of feature correlations, work on transfer learning in logistic classifiers is not new.

https://www.sciencedirect.com/science/article/abs/pii/S0167639314001010

“For each of the four combinations of MNIST and SVHN, we sample 60,000 images for testing and resample 60,000 images for each epoch in training.” Can you clarify the resampling here? Resampling with replacement might lead to data leakage.

There seems to be a major asymmetry in performance on SVHN-MNIST and MNIST-SVHN, which initially suggests that Alice can learn more useful features from SVHN for MNIST than the other way around. However, it is surprising that MNIST-SVHN can outperform SVHN-SVHN for the fully connected network, and the accuracy of MNIST-MNIST initially drops until SVHN-SVHN matches it for the rest of the values of β for the convolutional network. This implies that there may be other important unexplained factors neglected in the analysis.

It is well-established that progressively fine-tuning deeper layers tends to improve performance, up to the point where the model begins to overfit or saturate the dataset’s capacity. Additionally, low correlation of features naturally amplifies the differences in performance across fine-tuning strategies. It’s unclear what novel insight the authors intended to highlight in Figure 5.

Is it possible to extend the oracle-based correlation analysis using feature similarity metrics like Singular Vector Canonical Correlation Analysis (SVCCA)?

https://proceedings.neurips.cc/paper_files/paper/2017/file/dc6a7e655d7e5840e66733e9ee67cc69-Paper.pdf

The paper’s definition of “task similarity” is conflated with two other existing concepts distinct from cross-task transfer learning. MNIST to SVHN or vice versa is an example of source-free domain adaptation. MNIST to MNIST or SVHN to SVHN are evaluations within the same domain.

https://arxiv.org/pdf/2302.11803

Improved model performance due to spurious correlations, like background location, is an example of shortcut learning.

https://arxiv.org/abs/2412.05152

The paper does not clearly situate itself within the broader landscape of transfer learning, domain adaptation, and shortcut learning, resulting in conceptual ambiguity. Additionally, it lacks engagement with established benchmarks, methodologies, and recent literature essential for contextualizing its contributions and articulating its novelty.

Comments:

While the paper raises a few interesting points, its overall contribution is undermined by a lack of engagement with the surrounding literature due to a conflation of key terminology. This conceptual confusion led to methodological oversights, such as the neglect of possible confounding factors and the use of benchmarks that do not align with the stated goals. I encourage the authors to clarify their theoretical framing and resubmit to a workshop.

**Score:**

2

**Topic Fit:**

3

---

### Official Review · Reviewer_zRkg · 2025-09-15
**Paper review**

**Confidence:** 3

**Review:**

In this paper, the author investigate how feature and task correlation influences transfer learning. They study different scenarios, including a completely synthetic task (defined by binary feature vectors), a more complex visual task (where features are concatenations of images from two different datasets, MNIST and SVHN), and real-world image data (where the correlation between images is defined based on some speculative relation between the inputs).

In the synthetic scenario, larger correlation between pre-training and transfer tasks imply better performances (both in the case of pairwise and globally correlated features) with and without fine tuning. In the experiment on MNIST data, the authors find that pre-training can increase performance regardless of the correlation between tasks. In addition, they show additional results on interpretability (using attribution methods) and oracle feature selection (by zeroing out irrelevant features). Finally, the experiments on real-world images show that tasks that are semantically closer benefit to some extent of the pre-training initialization, while tasks that are not achieve around random chance.

**Strengths**

- The paper is overall well written and easy to follow. The experiments are well documented and the results are well-visualized and easy to interpret.

**Weaknesses**

- Some choices in the evaluation are not principled. For instance, in the experiments on real-world data, the definition of correlation is quite hand wavy and the actual correlation between speculated and actual properties should be quantitatively evaluated.
- While interesting, the results are not surprising and overall in line with "common sense" knowledge on transfer learning and feature/task correlation.
- No tests on the statistical significance of the results are included in the paper.

**Score:**

2

**Topic Fit:**

2

---

### Official Review · Reviewer_Zop6 · 2025-09-15
**paper review**

**Confidence:** 4

**Review:**

Summary: The paper investigates why reusing a pretrained network (used by Alice) sometimes boosts performance for a new downstream task (used by Bob). The paper studies a toy-feature correlation model and synthetic image datasets and finds that Bob's transfer accuracy increases with the correlation between Alice's and Bob's tasks/features. At zero correlation, the inductive biases of the model become more crucial for understanding performance as well as optimization artifacts. The authors further hypothesize that the optimal number of layers to freeze acts as a diagnostic for task/feature correlation.

Strengths
•	Well-designed test-bed: Algorithm 1 neatly isolates task correlation while keeping features spatially separable, enabling clean ablations
•	Empirical Results: Monotonic transfer vs beta (Fig. 4) and the "unfreeze more when less correlated” pattern are interesting and should be studied more in depth.
•	Zero-initializing connections to Bob only pixels demonstrates that leakage can be reduced and that some above-random transfer is optimization and architecture-driven.
Weaknesses
•	I think the monotonicity results are less surprising and require less emphasis. I would actually rewrite with more of a focus on the role of the architecture and unfreezing results. I think the fact that stronger task and feature correlations lead to better results is fairly obvious.
•	A methodological weakness, from what I can tell, is that correlation is defined via label coupling, not measured in features. Beta controls the probability that right labels match left labels but does not quantify mutual information between representations. This can confound “task correlation” with co-occurrence statistics and dataset construction artifacts. No quantitative correlation measure like CKA. I would measure this and report CKA as well.
•	Synthetic Setup: Horizontal concatenation creates a sharp vertical boundary and predictable spatial layout; convolution and weight sharing can propagate right side patterns into left trained features. This makes the Bob setup layout dependent as well. I would randomize the layout position to be left/right or, in fact, consider introducing vertical concatenation and masking.

Overall, I think this paper is reasonable and can be accepted. The weaknesses can be addressed in a fuller version of this paper.

**Score:**

4

**Topic Fit:**

2